# Mortality and Years of Potential Life Lost Due to COVID-19 in Brazil

**DOI:** 10.3390/ijerph18147626

**Published:** 2021-07-18

**Authors:** André Peres Barbosa de Castro, Marina Figueiredo Moreira, Paulo Henrique de Souza Bermejo, Waldecy Rodrigues, David Nadler Prata

**Affiliations:** 1Department of Strategic Articulation of Health Surveillance, Secretariat of Health Surveillance, Ministry of Health, Brasília 70719-040, Brazil; 2Faculty of Economics, Administration, Accounting and Information Science, University of Brasilia, Brasília 70910-900, Brazil; marinamoreira@unb.br (M.F.M.); paulobermejo@next.unb.br (P.H.d.S.B.); 3Institute of Regional Development, Graduate Program of Computational Modelling, Federal University of Tocantins, Palmas 77001-090, Brazil; waldecy@uft.edu.br (W.R.); ddnprata@uft.edu.br (D.N.P.)

**Keywords:** COVID-19, years of potential life lost, excess mortality

## Abstract

In November 2020, Brazil ranked third in the number of cases of coronavirus disease 2019 (COVID-19) and second in the number of deaths due to the disease. We carried out a descriptive study of deaths, mortality rate, years of potential life lost (YPLL) and excess mortality due to COVID-19, based on SARS-CoV-2 records in SIVEP-Gripe (Ministry of Health of Brazil) from 16 February 2020, to 1 January 2021. In this period, there were 98,025 deaths from COVID-19 in Brazil. Men accounted for 60.5% of the estimated 1.2 million YPLLs. High YPLL averages showed prematurity of deaths. The population aged 45–64 years (both sexes) represented more than 50% of all YPLLs. Risk factors were present in 69.5% of deaths, with heart disease, diabetes and obesity representing the most prevalent comorbidities in both sexes. Indigenous people had the lowest number of deaths and the highest average YPLL. However, in indigenous people, pregnant women and mothers had an average YPLL of over 35 years. The excess mortality for Brazil was estimated at 122,914 deaths (9.2%). The results show that the social impacts of YPLL due to COVID-19 are different depending on gender, race and risk factors. YPLL and excess mortality can be used to guide the prioritization of health interventions, such as prioritization of vaccination, lockdowns, or distribution of facial masks for the most vulnerable populations.

## 1. Introduction

In December 2019, cases of a new coronavirus were reported in Wuhan, China [1,2], called severe acute respiratory syndrome severe coronavirus 2 (SARS-CoV-2) or coronavirus disease 2019 (COVID-19) [3]. On 22 January 2020, the World Health Organization (WHO) confirmed person-to-person transmission of disease [4]. On January 31, because of the spread of COVID-19, the WHO declared a global outbreak and, on March 11, recognized it as a pandemic [5,6].

The first case in Brazil was registered on 26 January 2020 and a public health emergency was declared on February 3. In the 11th epidemiological week (EW), more than 100 cases were reported and, in the 12th EW, more than 1000 cases were reported. Community spread of COVID-19 was recognized in Brazil on March 20 [7,8]. In the 41st EW, Brazil was ranked third in cases reported (5,566,049) and second in the number of deaths (160,496) [9].

Since the first cases, COVID-19 has caused a significant number of deaths worldwide [4] and has been recognized as an important public health challenge because of human, economic and material losses [3,6]. Acter et al. (2020) explained that the pandemic is a health, humanitarian and development crisis that is advancing worldwide, causing deaths, damaging immune systems and weakening economies worldwide [3].

Hospitalized cases of severe acute respiratory syndrome (SARS) must be reported in the Epidemiological Surveillance System of Influenza (SIVEP-Gripe). These records identify cases and deaths related to Influenza A and B, Respiratory Syncytial Virus, Adenovirus and Parainfluenza 1, 2 and 3. In 2020, SIVEP-Gripe began to include cases of hospitalization and all deaths from COVID-19. In the 12th EW, it began including the results of RT-PCR tests in the records. The first recorded death was during the ninth EW [8,10].

The indicator of years of potential life lost between 1 and 70 (YPLL), used in mortality analyzes, helps in health planning, the identification of priorities and the assessment of social costs for premature deaths [11], although it does not measure morbidity or disability from an illness [12]. By expressing the social damage caused by premature death from COVID-19, this indicator provides an alert to society and supports the formulation of public policies to combat disease.

Excess mortality refers to the total number of deaths that occur during a period above what is expected for that period. In a pandemic, the number of deaths tends to rise not only because of the disease but also because of “collateral damage,” such as deaths from other health conditions that have not been adequately treated because of the overloaded health system. Excess mortality is an important social indicator of the social and economic impacts and consequences of a pandemic [13,14].

The objective of this work was to analyze, in an exploratory way, the impact of COVID-19 in Brazil by analyzing the data on deaths and potential years of life lost. In addition, excess mortality in Brazil during the year 2020 is presented as an indicator of the epidemiological scenario during the pandemic period. The hope is that the results of this study can contribute to other analyses concerning COVID-19, allowing future analyses and comparisons of the direct and indirect impacts of the pandemic in Brazil with those of other countries or territories.

## 2. Materials and Methods

Descriptive data on deaths with symptom onset dates between EW 8 and 53 of 2020 (from 16 February 2020 to 1 January 2021) were recorded in SIVEP-Gripe and made available for public use after anonymizing the records [15]. In this study, we calculated death rates, YPLLs due to COVID-19 in Brazil and excess mortality in 2020. 

### 2.1. SIVEP-Gripe

For notification in SIVEP-Gripe, all cases or deaths from SARS were considered, regardless of hospitalization, that presented influenza syndrome characterized by at least two of the following signs and symptoms: fever (even if referred), chills, sore throat, headache, cough, runny nose, smell disturbance and taste disturbance and those who also presented with dyspnea/respiratory discomfort, persistent chest pressure, O_2_ saturation of less than 95% in room air, or bluish coloration of the lips or face [10].

We used the SIVEP-Gripe database updated on 18 January 2021. Cases in which sex was ignored and those with ages below 1 year or above 70 years were not included. Three new variables were created: age groups with five-year intervals (except 1–4 years), YPLL and comorbidity. Other variables recorded were race/color, to create the category “Black” (cases registered as black and brown) and the “final classification” of the case, which was based on the reclassification of cases registered as the final classification “other” but with a description matching COVID-19 (Figure A1).

Mortality rates for Brazil and its geographic regions, proportional mortality, estimated YPLL by sex, age group, education, race/color and comorbidity and the YPLL rate by region and federation were calculated. For the denominator of the rates, we used the population estimated by IBGE for the year 2020 [16]. Analyses involving risk factors are presented separately, since each death was recorded with some comorbidity and may present one or more of these factors.

### 2.2. Estimates of Years of Potential Life Lost 

The YPLL method, proposed by Romeder and McWhinnie (1977), is the sum of the number of deaths at each age between 1 and 70 years old, multiplied by the remaining years of life up to 70 years old, organized into age groups with intervals of 5 years and as-summing a uniform distribution of deaths in the age groups [12]. We calculated the YPLL using Equation (1), where *ai* corresponds to the difference between the upper limit age considered (n) and the age of occurrence of death and *di* represents the total number of deaths that occurred at that age:(1)$$YPLL=\overset{n}{\underset{i=1}{\sum }}(a_{i}d_{i})$$

#### 2.2.1. Rate of Years of Potential Life Lost 

To compare YPLL between populations of different sizes, the rate of YPLL (YPLL rate) per 100,000 inhabitants can be used [12], where N is the number of inhabitants in the population between the ages considered (Equation (2)):(2)$$YPLL\;rate=\overset{n}{\underset{i=1}{\sum }}a_{i}d_{i}\times \frac{100,000}{N}$$

#### 2.2.2. Ratio of Years of Potential Life Lost

The COVID-19 YPLL ratios were calculated by gender and for other causes of death from respiratory and circulatory diseases and from land transport accidents. For other causes of death, we selected Brazilian Mortality Information System (SIM) records [17] that listed the following causes of death, according to the 10th revision of the International Classification of Diseases (ICD-10): diseases of the respiratory system (J00–J99), diseases of the circulatory system (I00–I99) and land transport accidents (V01–V89). Data were recorded by state and age between 1 and 70 years old [18].

### 2.3. Estimate of Excess Mortality

Excess mortality is defined as the sum of the differences between expected and observed deaths from all causes for each month in a specific year. For this study, we used the model proposed by Vital Strategies in collaboration with WHO [19]. For the projection of expected deaths for the year 2020, we used SIM data [17] from the period 2015–2019 to calculate the monthly historical averages and the lower and upper limits. 

Total deaths for the year 2020 were obtained from the Civil Registry Transparency Portal of the Civil Registry Information Center (CR) [20], as SIM data for the year 2020 were not available. Although the CR presents an under-registration, the correction factors made available by the Brazilian Institute of Geography and Statistics (IBGE) referred to the year 2018 [21], so we chose not to apply the correction factor.

From the monthly historical mean and using the standard error, confidence intervals (CIs) were calculated, with the lower and upper values corresponding percentile 2.5 and 97.5. Excess mortality is the sum of the monthly difference between the expected deaths in the period (upper limit) and the total number of deaths observed in the same period.

### 2.4. Data Analysis

The YPLL mean and percentage were stratified by sex, age group, race/color, zone, risk factor and pregnancy status for women, as were the mortality rate, YPLL average and YPLL rate by region and states of residence. The analyses were performed with SPSS 19.0 (IBM Corp, Armonk, NY, USA).

## 3. Results

### 3.1. Mortality Rate

Of the deaths, 59,608 (60.8%) were male, 44,134 (45.0%) were black and 33,278 (33.9%) were white. Regarding education, 11,790 (12.0%) had between 10 and 12 years of schooling and 10,564 (10.8%) had between 1 and 5 years of schooling. Individuals aged between 60 and 69 years represented 45,467 (46.4%) of deaths, followed by those between 50 and 59 years, who represented 25,344 deaths (25.9%). Among fatalities, 73.0% involved a risk factor (Table 1).

The geographic region of the country with the highest number of recorded deaths from COVID-19 was the south-east, with 46,003 cases (46.9%), followed by the north-east, with 22,211 cases (22.7%). Most deaths occurred among urban area residents, with 82,014 deaths (83.7%).

During the period, the COVID-19 mortality rate in Brazil was 46.3 deaths/100,000 inhabitants (Table A1). The northern region presented a mortality rate of 53.7, followed by the south-eastern and mid-western regions, with mortality rates of 51.7 and 50.7, respectively. The north-eastern and southern regions had lower mortality rates (Figure 1).

### 3.2. Years of Potential Life Lost

The estimated YPLL was 1,280,839, with the south-east and north-east accounting for 577,128 and 313,004 YPLLs, respectively (combined accounting for 69.5% of YPLLs). The average YPLL was 13.1 years for each COVID-19-related death, with the northern and north-eastern regions averaging over 14 years for each death. In the northern, mid-western and south-eastern regions, the YPLL was higher than the calculated rate of 604.9 years per 100,000 population in Brazil as a whole (Table A2).

The states with the greatest number of deaths were São Paulo, Rio de Janeiro and Minas Gerais in the south-east and Pernambuco and Ceará in the north-east, accounting for 53,478 deaths (54.6%). Of the 27 states, 12 had mortality rates above the national rate. The five states with the highest rates were Rio de Janeiro (78.7), Amazonas (78.5), Distrito Federal (68.2), Roraima (61.5) and Sergipe (59.5), as shown in Figure 1.

São Paulo, Rio de Janeiro, Minas Gerais, Pernambuco and Ceará contributed 53.2% of the total YPLLs over the study period. The mean YPLL for eight states was lower than the national average. The five states with the highest averages were Amapá (15.6), Roraima (15.4), Acre (15.3) and Rondônia (14.9), all in the northern region. YPLL rates of 15 states were higher than the national average. Amazonas (1083.3), Rio de Janeiro (1,001.2), Roraima (944.6), Sergipe (910.1) and the Federal District (890.1) were the states with the highest YPLL rates (Figure 1).

Regarding race/color, 47.5% of the YPLLs occurred in blacks, remaining above 47% when examined by gender. Indigenous people had a higher average YPLL (16.8 years). In indigenous women, this average was 19.2 years per death, 3.7 years more than for males.

Of women’s deaths, 227 occurred in pregnant women, with a total of 8,406 years lost and an average of 37.0 years. Urban areas represented 83.2% of the estimated YPLLs, while the highest average YPLL (14.3) was observed for rural residents. 

People aged 5–9 and 65–69 years had the lowest (90) and highest (25,114) number of death records, respectively. Those aged 45–64 years represented 55.5% of the estimated YPLL, whereas those aged 55–59 years accounted for 15.0% of the total YPLLs. The average YPLLs per death decreased with age, with a higher mean YPLL among those younger than 9 years (Figure 2).

Of the YPLLs, 889,771 (69.3%) occurred among those who had a risk factor. Women with risk factors accounted for 74.1% of the YPLLs, with an average of 12.7 years for each death. Among the deaths with risk factors, chronic cardiovascular disease and diabetes mellitus represented 365,541 and 335,956 YPLLs, respectively (Figure 3). It should be noted that a single death record can be associated with more than one risk factor.

The YPLL ratio between the genders showed that the YPLL was higher for men in all states, with 15 states showing higher values than the nationwide value (1.541) (Figure 4A). The ratio of YPLLs due to COVID-19 vs. other respiratory diseases that occurred in 2019 showed that the YPLL was lower for COVID-19 only for the state of Amapá and 10 states had a higher ratio than that of Brazil overall, which was 1.808 (Figure 4B). Based on the ratio of fatalities due to COVID-19 vs. terrestrial transport accidents, nine states had a higher YPLL for accidents than for COVID-19, eight states had lower ratios than the overall ratio of Brazil (1.376) and 10 had ratios above 1.375 (Figure 4D). In five Brazilian states, the YPLL ratio of deaths due to COVID-19 vs. other cardiovascular diseases was higher than 1; in the other states, the ratio was lower than 1, along with the nationwide ratio (0.609), showing that heart diseases had a higher YPLL in 2019 than COVID-19 in 2020 (Figure 4C). The state of the Amazonas was the largest contributor of COVID-19-related YPLLs and other causes of mortality, sometimes reaching values nine times higher than those calculated for the country.

### 3.3. The Expected and Excess Mortality Rate

Excess mortality in Brazil for the year 2020 was calculated at almost 123,000 deaths (9.2%). São Paulo and Rio de Janeiro, both in the south-east, had the highest excess mortality. In other regions where the states of the Amazon (north), Pernambuco (north-east), Distrito Federal (mid-west) and the Rio Grande do Sul (south). The state of Ceará (north-east) had the highest percentage (218.4%) of excess mortality, followed by the Federal District (mid-west), with 43.6% (Table A1). Excess mortalities in São Paulo and Rio de Janeiro (south-east) were among the highest percentages, both around 19%. The state of Piauí (north-east) did not present any excess mortality in the evaluated period (Figure 5).

The south-east and the south had the highest values of excess mortality, but the mid-western and south-eastern regions had the highest percentages (13%). In the north, North-east and south-east, the increase in fatalities began in May and, in the mid-west and south, the increase began in July (Figure 5).

## 4. Discussion

During the study period, more than 1.2 million YPLLs were recorded in Brazil. The YPLLs due to COVID-19 in 2020 was higher than the YPLLs for land transport accidents in 2013, estimated at 1,309,191.5 years by Andrade and Mello-Jorge (2016); YPLLs due to COVID-19 were also greater than the 608,059 YPLLs estimated by Bochner and Freire (2020) due to deaths from poisoning during the period between 2010 and 2015 [22,23]. Transport accidents and intoxications have a greater impact on people aged 20–49 years. The difference in YPLLs found between these two causes occurred because of the higher number of deaths due to COVID-19.

Perea et al. (2018) estimated 1,589,501 YPLLs due to deaths from oral and pharyngeal cancer from 1979 to 2013, in Brazil. The YPLLs due to COVID-19 corresponded to 80.6% of the YPLLs for these types of cancer during this 35-year historical period [24].

Race/color disparities in both COVID-19 incidence and mortality have affected the black population more [25,26]. The differences in deaths and YPLLs between blacks and whites reveal distinct impacts of COVID-19 on these populations. Baqui et al. (2020) analyzed hospitalized cases (in SIVEP-Gripe) and reported that deaths among browns and blacks were higher than in whites, indigenous people and Asians; in the states of the north and north-east, this difference was more than 40% and, in the other states, the difference was more than 29% [27]. Pierce et al. (2020) analyzed COVID-19 mortality in Chicago (USA) and estimated the YPLLs per 100,000 inhabitants at 559 years for the black population and 312 years for whites [28]. 

These findings were discussed as a reflection of racial inequalities by Wrigley-Field (2020), who predicted that the difference in life expectancy between whites and blacks will be even greater because of the COVID-19 pandemic [29]. Oliveira et al. (2020) pointed out that racial inequalities and structural racism in Brazil should be considered when assessing the higher lethality among hospitalized blacks than among whites, although the number of hospitalized blacks was also higher [30].

The gender differences related to COVID-19 are evidenced by the number of deaths, total YPLLs and the ratio between the sexes, with predominance in men for all these indicators. In our study, men accounted for 53% more YPLLs than women and Arolas et al. [31] found a 44% percentage in a study conducted in 81 countries. These authors cited a higher mean age among women, resulting in fewer YPLLs and a higher number of deaths in absolute values among men, possibly explaining the disparities between our findings. The YPLL in indigenous people was lower, but the average number of years lost was higher than the overall average, with more than 19 years for females, indicating a higher impact on younger age groups.

However, the literature indicates that deaths in pregnant and postpartum women result from the evolution of the disease and the presence of comorbidities [32,33,34]. SIVEP-Gripe has no records of outcomes in pregnant women, such as abortions, stillbirths, or vertical transmission of COVID-19, which represent social losses not accounted for in YPLL, although this is discussed in the literature [35,36]. The YPLL of pregnant and postpartum women found is due to the fact that most women in Brazil become pregnant between the ages of 20 and 44 [37].

Abate et al. (2021) conducted a systematic review and found a 48% prevalence of comorbidities in hospitalized patients, with diabetes mellitus (48%) predominating, followed by hypertension and cardiovascular diseases (15%) [38]. Among the main risk factors listed in this study were chronic cardiovascular disease, diabetes mellitus and obesity, which have also been found in other studies [39,40,41,42]. Our findings indicated the presence of a risk factor in 73% of deaths, but these results can be explained by the fact that we only examined the number of deaths. The literature has shown a correlation of obesity and other comorbidities with a higher prevalence of hospitalizations and mortality in COVID-19 infected patients. However, more specific analyses on the presence of risk factors could not be performed because of the high number of records for which this information was either ignored or missing (Table A3) [39,41,43].

Excess mortality is an easy-to-calculate indicator; by using up-to-date data and YPLL to assess and track the evolution of COVID-19, this indicator helps in understanding how the disease affects populations and territories. It is important to note that excess mortality is not only a direct reflection of COVID-19, but is also attributable to deaths not related to the pandemic because of the overloading of health services and failures in the treatment of chronic diseases [44].

## 5. Conclusions

In 2020, almost one year after the first case of COVID-19 in Brazil, more than one million YPLLs were reported, more than half of which were due to deaths in the economically active population. Premature mortality reduces productivity and increases costs for the national health care system, including costs of hospitalization and treatment before death and social security costs due to payments to dependents, raising the economic impact on society. 

YPLL rates allow us to compare the dynamics of COVID-19 in Brazilian states, even though they have heterogeneous socioeconomic and geographic characteristics. The state of Amazonas in the northern region of the country and Rio de Janeiro in the southeastern region had YPLL rates above 1000 years of life lost per 100,000 inhabitants, showing that the social losses in these states were quite similar, even though the total number of deaths differed.

The excess mortality identified during the period, although not directly associated with COVID-19 deaths, shows that mortality increased as the pandemic progressed. The increase in mortality can also be explained by the overloading of health services and the difficulties of the health system to restructure itself to face the challenges of a pandemic. In this scenario, other diseases and illnesses were also affected; they no longer received proper care and monitoring because of the lack of available outpatient services, hospital beds and health professionals, as these services were displaced to respond to the pandemic. 

The deaths of pregnant or postpartum women not only represent a high social cost but also leave hidden, unmeasurable social losses, such as possible abortions or stillbirths. Similarly, COVID-19 deaths among the indigenous population carry a high social burden, especially for ethnicities with smaller and more vulnerable populations. As such, additional studies to understand these unmeasurable impacts are needed.

The monitoring of these indicators helps in the planning of health actions that seek to reduce early mortality, such as the definition of more vulnerable groups that should receive special attention, including priority groups for vaccination. 

This study has limitations regarding the lack of data for some variables, such as education, race/color and risk factors (Table A4). These percentages may have caused bias in the results. Since education had the highest percentage of ignored information, it was not considered for the YPLL evaluations. YPLL rates may have a bias because they used data from the year 2019 on other causes of mortality because of the absence of this information for the year 2020. The excess mortality calculations may be underestimated because of the decision not to use correction factors, as they represent delays in death records for SIM data. This may also explain the absence of excess mortality in the state of Piauí.

## Figures and Tables

**Figure 1 ijerph-18-07626-f001:**
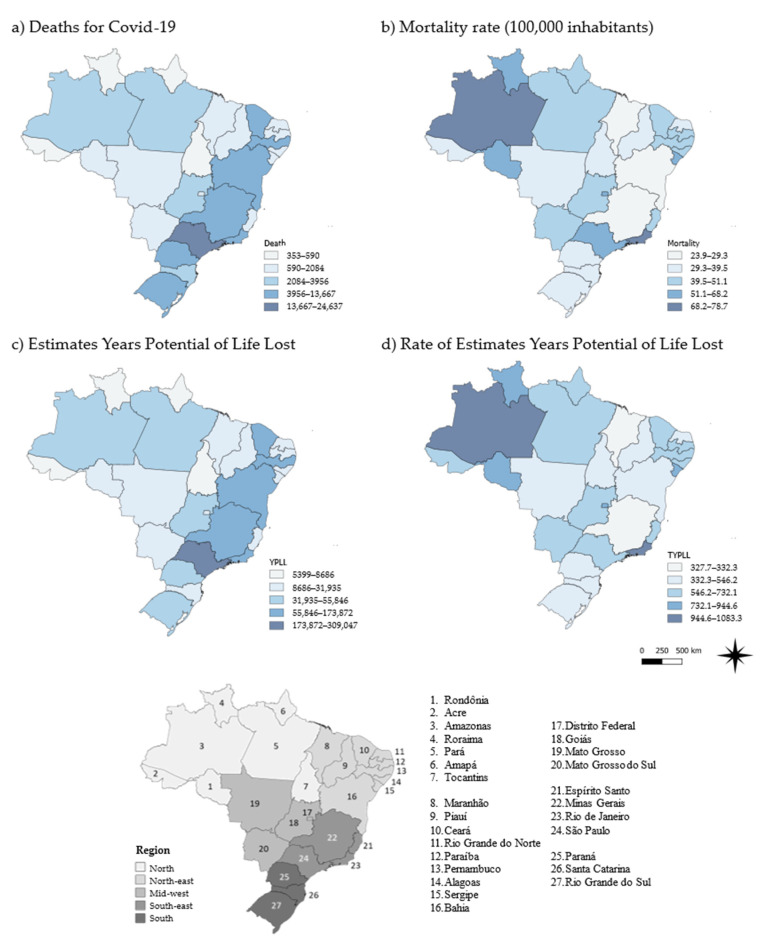
Fatalities and years of potential life lost due to COVID-19 in Brazil (2020) by region and state of residence.

**Figure 2 ijerph-18-07626-f002:**
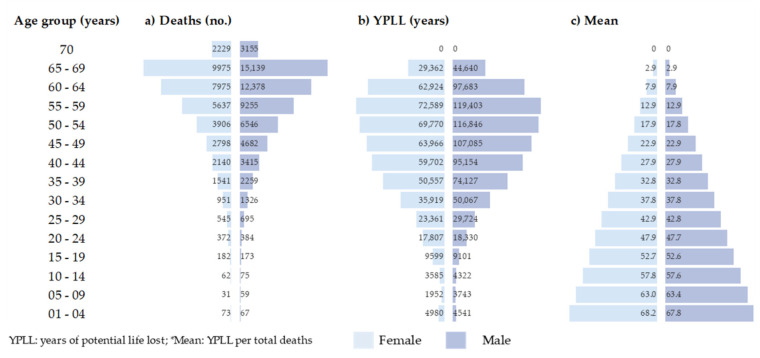
Deaths and years of potential life lost due to COVID-19 in Brazil (2020) by gender and age group.

**Figure 3 ijerph-18-07626-f003:**
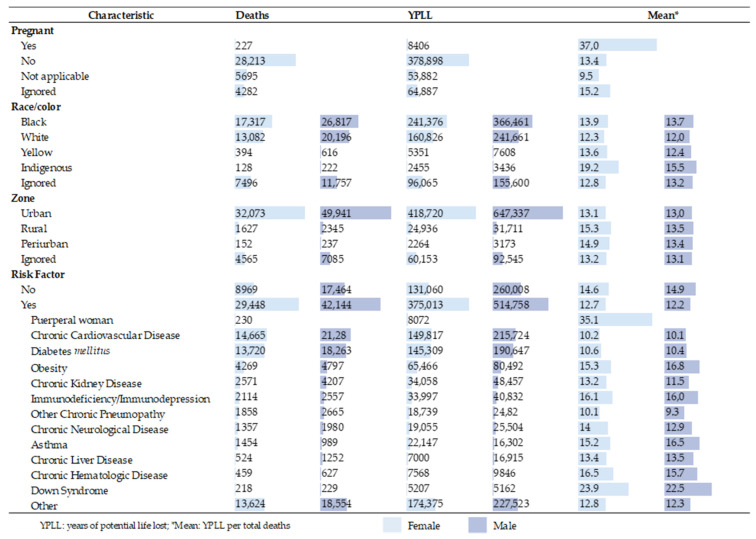
Fatalities and years of potential life lost in Brazil (2020) due to COVID-19, by gender, race/color, zone residence and risk factor.

**Figure 4 ijerph-18-07626-f004:**
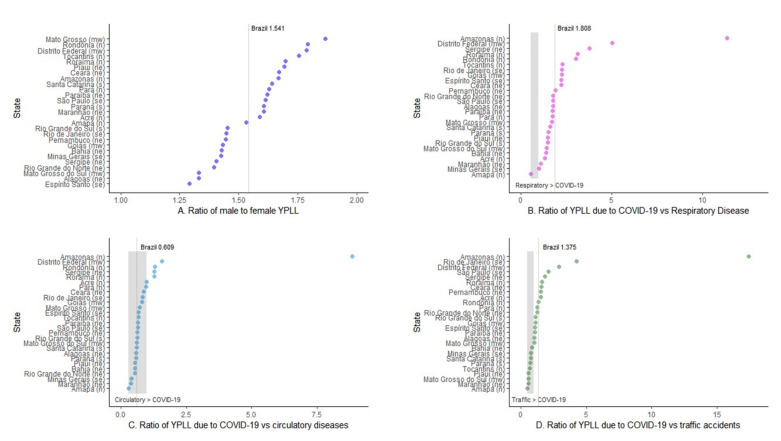
Years of potential life lost (YPLL) ratio in Brazil (2020) by sex (**A**), other respiratory diseases (**B**), heart disease (**C**) and land traffic accidents (**D**). When mortality, by sex or other causes, affects the YPLL equally, the ratio is 1. In charts **B**–**D**, values above 1 mean that the YPLL for COVID-19 was higher than for the other causes, the opposite is true for values less than 1. The Brazil-wide ratio is represented by the vertical line on each chart. The names of the states followed by the letters (in parentheses) represent the region of the country in which they are located (n = north, ne = north-east, mw = mid-west, se = south-east and s = south).

**Figure 5 ijerph-18-07626-f005:**
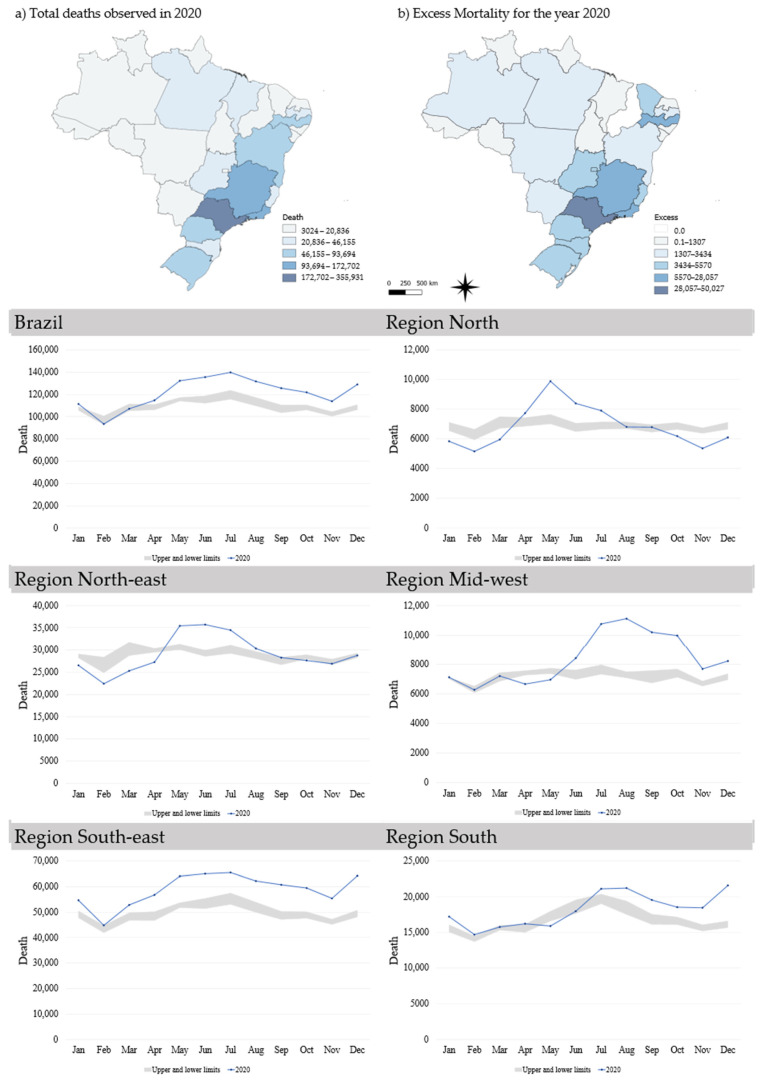
Excess mortality in Brazil (2020). Deaths per month compared with upper and lower limits (95% CI) of historical averages in Brazil and its regions.

**Table 1 ijerph-18-07626-t001:** Distribution of SARS deaths due to COVID-19 in Brazil (2020) and proportional description according to sociodemographic characteristics.

Characteristics	Female, no. (%)	Male, no. (%)	Total, no. (%)	*X*^2^ Square	df	*p*
	38,417 (39.2)	59,608 (60.8)	98,025 (100)			
**Race/color**						
Black	17,317 (45.1)	26,817 (45.0)	44,134 (45.0)	1.796	4	0.773
White	13,082 (34.1)	20,196 (33.9)	33,278 (33.9)			
Yellow	394 (1.0)	616 (1.0)	1010 (1.0)			
Indigenous	128 (0.3)	222 (0.4)	350 (0.4)			
Ignored	7496 (19.5)	11,757 (19.7)	19,253 (19.6)			
**Education (years)**						
No schooling/Illiterate	1082 (2.8)	1230 (2.1)	2312 (2.4)	206.919	6	0.000
1–5 years	4466 (11.6)	6098 (10.2)	10,564 (10.8)			
6–9 years	3304 (8.6)	5126 (8.6)	8430 (8.6)			
10–12 years	4288 (11.2)	7502 (12.6)	11,790 (12.0)			
>12 years old	1446 (3.8)	2896 (4.9)	4342 (4.4)			
Not applicable	44 (0.1)	41 (0.1)	85 (0.1)			
Ignored	23,787 (61.9)	36,715 (61.6)	60,502 (61.7)			
**Age Group**						
01–04	73 (0.2)	67 (0.1)	140 (0.1)	138.169	14	0.000
05–09	31 (0.1)	59 (0.1)	90 (0.1)			
10–14	62 (0.2)	75 (0.1)	137 (0.1)			
15–19	182 (0.5)	173 (0.3)	355 (0.4)			
20–24	372 (1.0)	384 (0.6)	756 (0.8)			
25–29	545 (1.4)	695 (1.2)	1240 (1.3)			
30–34	951 (2.5)	1326 (2.2)	2277 (2.3)			
35–39	1541 (4.0)	2259 (3.8)	3800 (3.9)			
40–44	2140 (5.6)	3415 (5.7)	5555 (5.7)			
45–49	2798 (7.3)	4682 (7.9)	7480 (7.6)			
50–54	3906 (10.2)	6546 (11.0)	10,452 (10.7)			
55–59	5637 (14.7)	9255 (15.5)	14,892 (15.2)			
60–64	7975 (20.8)	12,378 (20.8)	20,353 (20.8)			
65–69	9975 (26.0)	15,139 (25.4)	25,114 (25.6)			
≥70	2229 (5.8)	3155 (5.3)	5384 (5.5)			
**Risk Factor**						
No	8969 (23.3)	17,464 (29.3	26,433 (27.0)	420.173	1	0.000
Yes	29,448 (76.7)	42,144 (70.7)	71,592 (73.0)			
**Region of Residence**						
North	3674 (9.6)	6345 (10.6)	10,019 (10.2)	35.475	4	0.000
Northeast	8901 (23.2)	13310 (22.3)	22211 (22.7)			
Midwest	3287 (8.6)	5081 (8.5)	8368 (8.5)			
Southeast	18,121 (47.2)	27,882 (46.8)	46,003 (46.9)			
South	4434 (11.5)	6990 (11.7)	11,424 (11.7)			
**Zone**						
Urban	32,073 (83.5)	49,941 (83.8)	82,014 (83.7)	5.473	3	0.140
Rural	1627 (4.2)	2345 (3.9)	3972 (4.1)			
Periurban	152 (0.4)	237 (0.4)	389 (0.4)			
Ignored	4565 (11.9)	7085 (11.9)	11,650 (11.9)			

## Data Availability

Public data are anonymized and made available by the Brazilian Government. Data from the SARS Surveillance—COVID-19 are available at https://opendatasus.saude.gov.br/dataset/bd-srag-2020, accessed on 29 September 2020. Population estimates for Brazilian municipalities are available online at https://www.ibge.gov.br/estatisticas/sociais/populacao/9103-estimativas-de-populacao.html?=&t=downloads, accessed on 14 October 2020 and the Brazilian Mortality Information System (MIS) at http://www2.datasus.gov.br/DATASUS/index.php?area=0205&id=6937, accessed on 6 April 2021.

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
