# Peer review of "Mortality and Years of Potential Life Lost Due to COVID-19 in Brazil"

_ijerph, 2021, doi:10.3390/ijerph18147626_

Round 1
Reviewer 1 Report
This paper describes the excessive mortality and years of potential life lost by COVID-19 in Brazil. The time relevancy and dataset of the paper seems to be the two prominent strengths of the work as both researchers and public have a temporal interest in the topic. Particularly, the findings are supported by the huge and contemporary dataset which is used in this study. I have the following points related to each part of the paper, that the authors may like to consider for improving the paper and to enhance the readability and hence impact of the work.
- In the abstract, the authors have provided key and useful statistics related to the mortality and years of potential life lost in Brazil. The utility of these statistics can be enhanced if the authors can elaborate more on one of implications: “YPLL and excess mortality help in choosing appropriate health measures for vulnerable populations”. The notion of “appropriate” is not quite clear. The readers will be interested in some inferences and policy guidelines that the research may convey to manage some aspects of the pandemic better.
- In the introduction, the authors have introduced the topic and key terms very well and in brief words. It communicates the intention quite well. However, the authors seem not taking any benefit from the literature on excessive mortality and years of potential life lost. There are references to the statistics and other factual information, but the theoretical bases are not broad enough in the introduction to establish a broader thinking. These broad thoughts may be beyond the practitioners concrete focus and would therefore be more helpful. It is therefore important that the authors may bring some mortality theory or law of large numbers to create more probable expectations for interreferences for more useful theoretical and policy outcomes. I suggest the authors add a full section on related theory, right after introduction and before methods and material. This will also be important because the theory can provide useful conceptual categories for the data to be looked at, as currently only group and descriptive categories are made with little relevance to theoretical and policy concerns.
- Down to the line in materials and methods, I wonder why the authors have only considered descriptive statistics. As both the design and data are very much suitable for more developed methods for group differences and inferential findings. I am not suggesting the authors to change their methods, but a few lines of justifying the sufficiency of current descriptive quantities could be required. This is because most of what the data processing has been done (e.g., region and age wise deaths etc) are also done by even the newspapers and websites. The research has therefore an opportunity to use more rigorous analysis methods and thus bring more novel numbers as outputs. Perhaps, Chi-Square tests for group differences and linear programming for setting up constraints can be used to find the optimized levels of excessive mortality and potential years of life lost.
- In results, there are overall excessive tabular presentations are used except for two occasions, when the data is presented graphically and those make more sense than those overloaded tables. Since the data is in huge numbers and comparative, I guess more comparative infographics will be helpful to present the data comparatively through colour codes for example. More space needs to be spared for the authors commentary, and the tables could be kept in the appendix.
- The discussion section again presents the data and is therefore very descriptive. Discussion is usually expected to be discursive. A few studies are mentioned for the comparisons though. There is, nevertheless, an opportunity to derive more generalisable inferences out of the very valuable descriptive statistics if they are compared more rigorously with the theoretical debates. I would, therefore, suggest the authors to elaborate more on those related studies in other contexts and synthesis the findings of the paper with more generalisable inferences for recommendations. In this regard, if the authors introduce some theoretical insights in or right after introduction, then those theoretical insights will provide bases for creating a discussion afterwards.
- The conclusion section is very brief and summarising potential life lost by COVID-19. Initially the introduction was very optimistic about the outcomes to be related to data based measures related to the Covid -19. There are, however, little discussions and recommendations related what exact measures can be taken to reduce the potential life lost and other concerns related to the Covid-19, particularly in Brazil. As the paper has very rich dataset, I suggest the authors can at least come up with one key policy and practice implication related to each group perspective (e.g. gender, regions, race, and so forth). If more theory is included, there could be more possibility of making theory implications.
Finally, the paper is a very well-timing contribution and I wish the authors the best of luck. I suggest a little editorial attention to improvement of the language and presentation to increase the readability of the paper.
Author Response
Dear Reviewer,
We respond to the valuable comments made in our article.
Point 1 - In the abstract, the authors have provided key and useful statistics related to the mortality and years of potential life lost in Brazil. The utility of these statistics can be enhanced if the authors can elaborate more on one of implications: “YPLL and excess mortality help in choosing appropriate health measures for vulnerable populations”. The notion of “appropriate” is not quite clear. The readers will be interested in some inferences and policy guidelines that the research may convey to manage some aspects of the pandemic better.
Response 1 - Adjusted for “YPLL and excess mortality can be used to guide prioritization for interventions health as prioritization of vaccination, lockdown, or distribution of the facial mask for the most vulnerable population.”
Point 2 - In the introduction, the authors have introduced the topic and key terms very well and in brief words. It communicates the intention quite well. However, the authors seem not taking any benefit from the literature on excessive mortality and years of potential life lost. There are references to the statistics and other factual information, but the theoretical bases are not broad enough in the introduction to establish a broader thinking. These broad thoughts may be beyond the practitioners concrete focus and would therefore be more helpful. It is therefore important that the authors may bring some mortality theory or law of large numbers to create more probable expectations for interreferences for more useful theoretical and policy outcomes. I suggest the authors add a full section on related theory, right after introduction and before methods and material. This will also be important because the theory can provide useful conceptual categories for the data to be looked at, as currently only group and descriptive categories are made with little relevance to theoretical and policy concerns.
Response 2 – We reviewed the section and adjusted meet the request within the article's proposal.
Point 3 - Down to the line in materials and methods, I wonder why the authors have only considered descriptive statistics. As both the design and data are very much suitable for more developed methods for group differences and inferential findings. I am not suggesting the authors to change their methods, but a few lines of justifying the sufficiency of current descriptive quantities could be required. This is because most of what the data processing has been done (e.g., region and age wise deaths etc) are also done by even the newspapers and websites. The research has therefore an opportunity to use more rigorous analysis methods and thus bring more novel numbers as outputs. Perhaps, Chi-Square tests for group differences and linear programming for setting up constraints can be used to find the optimized levels of excessive mortality and potential years of life lost.
Response 3 – We appreciate the comments and have chosen to adjust, without substantially changing the article. The objective of the article was to provide a descriptive account of Covid-19 deaths in Brazil, to allow a look at the social impacts of the disease.
Point 4 - In results, there are overall excessive tabular presentations are used except for two occasions, when the data is presented graphically and those make more sense than those overloaded tables. Since the data is in huge numbers and comparative, I guess more comparative infographics will be helpful to present the data comparatively through colour codes for example. More space needs to be spared for the authors commentary, and the tables could be kept in the appendix.
Response 4 – We have complied with the request made by adding infographics and keeping the tables in the appendix.
Point 5 - The discussion section again presents the data and is therefore very descriptive. Discussion is usually expected to be discursive. A few studies are mentioned for the comparisons though. There is, nevertheless, an opportunity to derive more generalisable inferences out of the very valuable descriptive statistics if they are compared more rigorously with the theoretical debates. I would, therefore, suggest the authors to elaborate more on those related studies in other contexts and synthesis the findings of the paper with more generalisable inferences for recommendations. In this regard, if the authors introduce some theoretical insights in or right after introduction, then those theoretical insights will provide bases for creating a discussion afterwards.
Response 5 – We reviewed the section and adjusted meet the request within the article's proposal.
Point 6 - The conclusion section is very brief and summarising potential life lost by COVID-19. Initially the introduction was very optimistic about the outcomes to be related to data based measures related to the Covid -19. There are, however, little discussions and recommendations related what exact measures can be taken to reduce the potential life lost and other concerns related to the Covid-19, particularly in Brazil. As the paper has very rich dataset, I suggest the authors can at least come up with one key policy and practice implication related to each group perspective (e.g. gender, regions, race, and so forth). If more theory is included, there could be more possibility of making theory implications.
Response 6 – We have revised the section and adjusted the text to fit the comments of all the reviewers.
Point 7 - Finally, the paper is a very well-timing contribution and I wish the authors the best of luck. I suggest a little editorial attention to improvement of the language and presentation to increase the readability of the paper.
Response 7 – The article has been submitted for review to Cambridge Proofreading & Editing LLC (Order No. 935-98-15).
Reviewer 2 Report
The paper by Castro et al reports results on mortality and years of potential life lost by COVID-19 in Brazil. This study is potentially able to provide an informative overview over this specific aspect of the Covid-19 disease, however, the manuscript requires some revision.
Major points:
1) Language: The paper needs thorough proofreading and editing of grammar and wording
2) Discussion: The discussion consists mainly of citations of other studies with similar findings and interprets only marginally the own results. Please elaborate more on what the present study adds to the previous results and highlight how the results from Brazil are also of interest to an international audience of readers.
3) Section 2.3. in Methods needs to be revised. It is not clear how excess mortality is calculated here. For example, how were the deaths from the Civil Registry Transparency Portal merged or combined with the deaths from SIVEP-Gripe? What is meant by “without sub-record correction”? How are the numbers from the last column of table 2 (“No(%)”) calculated?
4) Results: The YPLL for each COVID-19-related death among pregnant women is reported as 37 years. However, this number is misleading since most women probably become pregnant between the age of 20 to 40. Therefore, the higher YPLL per death in this group reflects rather the younger age of pregnant women compared to the entire subgroup of females in the study than an increased risk of pregnant women.
Other points:
- Page 1, line 41: “On 41st EW, Brazil was third in cases reported […] [9].” The 41st EW of 2020 is in October, but reference [9] states that the data was downloaded in June 2020.
- Page 1, line 66: “[…] with symptom onset date between epidemiological weeks (SE) 08 to 53 […] recorded in SIVEP-Gripe”. a) the abbreviation for epidemiological week introduced was EW, not SE. Second, how were the cases of deaths for EW 8-11 analyzed if SIVEP-Gripe started to include cases “as of the 12th EW” (page 1, line 51)?
- Page3: equation (1) is not correct, it can be given as either “∑ (aidi)" or “(aidi) + (a[i+1] d[i+1]) + … + (an dn)”
- Table 1-4 headings: the data presented in the tables is from 2020, therefore the heading should read “Brazil, 2020” instead of “Brazil, 2021”.
- It seems a little odd to analyze the age group of 70 separately, whereas other ages are categorized in groups of 4 or 5 years. Please give the reason for this variable categorization.
- The full definition of the abbreviation TYPLL should be given at first mention.
Author Response
Dear Reviewer,
We respond to the valuable comments made in our article.
Point 1 - Language: The paper needs thorough proofreading and editing of grammar and wording.
Response 1 – The article has been submitted for review to Cambridge Proofreading & Editing LLC (Order No. 935-98-15).
Point 2 - Discussion: The discussion consists mainly of citations of other studies with similar findings and interprets only marginally the own results. Please elaborate more on what the present study adds to the previous results and highlight how the results from Brazil are also of interest to an international audience of readers.
Response 2 – The article seeks to show the impacts of Covid-19 for Brazil, by means of indicators that are known worldwide and that allow comparison with other countries. Today, Brazil has unfortunately attracted attention due to the high number of cases and deaths from Covid-19.
Point 3 - Section 2.3. in Methods needs to be revised. It is not clear how excess mortality is calculated here. For example, how were the deaths from the Civil Registry Transparency Portal merged or combined with the deaths from SIVEP-Gripe? What is meant by “without sub-record correction”? How are the numbers from the last column of table 2 (“No(%)”) calculated?
Response 3 – We have responded to the recommendations and suggestions by rewriting the methods section 2.3. The Civil Registry data is not merged with SIVEP-Gripe data because excess mortality is estimated for all causes of death.
Point 4 - Results: The YPLL for each COVID-19-related death among pregnant women is reported as 37 years. However, this number is misleading since most women probably become pregnant between the age of 20 to 40. Therefore, the higher YPLL per death in this group reflects rather the younger age of pregnant women compared to the entire subgroup of females in the study than an increased risk of pregnant women.
Response 4 – Thank you for your comment. We have reviewed our result and have chosen to exclude it from the article.
Point 5 - Page 1, line 41: “On 41st EW, Brazil was third in cases reported […] [9].” The 41st EW of 2020 is in October, but reference [9] states that the data was downloaded in June 2020.
Response 5 – The access date is wrong, the correct date is November 04, 2020. The reference has been corrected.
Point 6 - Page 1, line 66: “[…] with symptom onset date between epidemiological weeks (SE) 08 to 53 […] recorded in SIVEP-Gripe”. a) the abbreviation for epidemiological week introduced was EW, not SE. Second, how were the cases of deaths for EW 8-11 analyzed if SIVEP-Gripe started to include cases “as of the 12th EW” (page 1, line 51)?
Response 6 – Adjusted. In 2020, SIVEP-Gripe began to include cases of hospitalization and all deaths from COVID-19. On the 12th EW he included the RT-PCR in the notification form. The 1st recorded death on the 9th EW.
Point 7 - Page3: equation (1) is not correct, it can be given as either “∑ (aidi)" or “(aidi) + (a[i+1] d[i+1]) + … + (an dn)”
Response 7 – We made the correction in the equation.
Point 8 - Table 1-4 headings: the data presented in the tables is from 2020, therefore the heading should read “Brazil, 2020” instead of “Brazil, 2021”.
Response 8 – We made the correction in the tables and figures
Point 9 - It seems a little odd to analyze the age group of 70 separately, whereas other ages are categorized in groups of 4 or 5 years. Please give the reason for this variable categorization.
Response 9 – The method sets 70 years as the last upper age limit. Thus, we established only as information for readers of the profile of deaths by covid aged 70 years.
Point 10 - The full definition of the abbreviation TYPLL should be given at first mention.
Response 10 – We have revised and adjusted the text.
Reviewer 3 Report
The manuscript submitted by de Castro and coworkers is a very interesting contribution to the burden of COVID-19 in less developed countries. The scientific approach appears to be valid and the results are clearly presented. I have only two minor concerns that should be addressed in the Discussion section of the paper:
1.) The authors should state why they restricted the range of ages included in their analysis to 70 years. In Europe the highest proportion of deaths due to COVID-19 occur in the age groups of 60 and above. Are there no data available for higher age groups in the Brazilian statistics?
2.) The role of obesity as risk factor should further be discussed. According to the results of the paper obesity has only a moderate influence on deaths due to COVID-19 in Brasil. However, in the current literature worldwide obesity takes a prominent role as risk factor (e.g., see https://www.worldobesityday.org/assets/downloads/COVID-19-and-Obesity-The-2021-Atlas.pdf). How do the authors explain this discrepancy?
Author Response
Dear Reviewer,
We respond to the valuable comments made in our article.
Point 1 - The authors should state why they restricted the range of ages included in their analysis to 70 years. In Europe the highest proportion of deaths due to COVID 19 occur in the age groups of 60 and above. Are there no data available for higher age groups in the Brazilian statistics?
Response 1 – The method sets 70 years as the last upper age limit. Thus, we established only as information for readers of the profile of deaths by covid aged 70 years. We have records of deaths in all age groups, however, regarding the YPLL calculation method, we do not use ages above the upper limit of 70 years to avoid survival values, which would lead readers to imagine that Covid could have resulted in "years of lives gained".
Point 2 - The role of obesity as risk factor should further be discussed. According to the results of the paper obesity has only a moderate influence on deaths due to COVID-19 in Brasil. However, in the current literature worldwide obesity takes a prominent role as risk factor (e.g., see https://www.worldobesityday.org/assets/downloads/COVID-19-and-Obesity-The-2021-Atlas.pdf). How do the authors explain this discrepancy?
Response 2 – We have added a paragraph explaining why the findings differ from the literature in relation to obesity and other comorbidities. (Page 11 – line 291 – 295). “The literature has shown a correlation between obesity and other comorbidities with higher prevalence of hospitalizations and mortality in Covid-19 infected patients. However, more specific analyses for the presence of risk factors could not be performed because of the significant number of records in which this information was either ignored or blank.”.
Round 2
Reviewer 2 Report
Thank you for the opportunity to review the revised manuscript. It has improved a lot and the overall message of this analysis is now much clearer.
I have only some minor points of suggestion:
- Response 3 – We have responded to the recommendations and suggestions by rewriting the methods section 2.3.
- One small point here: the sentence in line 140/141 is a little confusing, but in general the lower and upper values of a 95% CI do not correspond to a 5% CI and 95% CI, but to the 2.5% and 97.5% quantile of the underlying distribution.
- Response 4 – Thank you for your comment. We have reviewed our result and have chosen to exclude it from the article.
- The results for pregnant women are still included in table 2 and page 6 line 193/194. However, these results also don't have to be excluded entirely from the manuscript, an additional explanation of potential confounding for this group added to the interpretation (in results and/or discussion) of YPLL per total deaths should be just fine.
-
Point 10 - The full definition of the abbreviation TYPLL should be given at first mention.
Response 10 – We have revised and adjusted the text.
- This is optional, but I am still curious what the "T" in the abbreviation TYPLL stands for.
Author Response
Dear Reviewer
Thank you for your comments on improving our article.
Response 3 – We have responded to the recommendations and suggestions by rewriting the methods section 2.3.
One small point here: the sentence in line 140/141 is a little confusing, but in general the lower and upper values of a 95% CI do not correspond to a 5% CI and 95% CI, but to the 2.5% and 97.5% quantile of the underlying distribution.
Response - We thank you for your comments and have rewritten the sentence to correct the information (page 3, lines 137 to 140):
“From the monthly historical mean, and using the standard error, confidence intervals (Cis) were calculated, with the lower and upper values corresponding percentile 2.5 and 97.5. Excess mortality is the sum of the monthly difference between the expected deaths in the period (upper limit) and the total number of deaths observed in the same period.”
Response 4 – Thank you for your comment. We have reviewed our result and have chosen to exclude it from the article.
The results for pregnant women are still included in table 2 and page 6 line 193/194. However, these results also don't have to be excluded entirely from the manuscript, an additional explanation of potential confounding for this group added to the interpretation (in results and/or discussion) of YPLL per total deaths should be just fine.
Response - Thanks for your comments and have added the following explanation to page 10, lines 281 to 283: “The YPLL of pregnant and postpartum women found is due to the fact that most women in Brazil become pregnant between the ages of 20 and 44.”
Point 10 - The full definition of the abbreviation TYPLL should be given at first mention.
Response 10 – We have revised and adjusted the text.
This is optional, but I am still curious what the "T" in the abbreviation TYPLL stands for.
Response - Thanks for the observation. The "T" referred to the abbreviation for "taxa" (rate) in Portuguese. We changed it to "YPLL rate" for better understanding.